# Realist evaluation of the impact of the research translation process on health system sustainability: a study protocol

Abby Mosedale [1] , Delia Hendrie [1] , Elizabeth Geelhoed,[2]
Yvonne Zurynski [3] , Suzanne Robinson [1]

¹School of Population Health, Curtin University, Perth, Western Australia, Australia
²School of Allied Health, University of Western Australia, Perth, Western Australia, Australia
³Australian Institute of Health Innnovation, Macquarie University, Sydney, New South Wales, Australia

**Correspondence to**
Abby Mosedale;
abby.mosedale@curtin.edu.au

## ABSTRACT

**Introduction** Sustainability at a system level relates to the capacity of the system to be able to service the ongoing health needs of the population. It is a multifaceted concept encompassing both the affordability and efficiency of a system and the system's ability to adapt and change. To address issues that currently threaten health system sustainability, healthcare leaders, policy makers, clinicians and researchers are searching for solutions to ensure the delivery of safe, value-based care into the future. The timely translation of research evidence into sustainable interventions that can be adopted into the health system is one way of bolstering the sustainability of the system as a whole. We present a protocol for the realist evaluation of a research translation funding programme to understand how the research translation process contributes to health system sustainability and value-based healthcare.

**Methods and analysis** Underpinned by the realist evaluation framework, we will: (1) Develop the Initial Program Theory (IPT) of the research translation process; (2) Test the program theory through case study analysis; and (3) Refine and consolidate the theory through stakeholder consultation. The evaluation uses a case example of a research translation programme, chosen for its representation of a microcosm of the broader health system and the heterogeneity of service improvement activities taking place within it. Across the three phases, analysis of data from documents about the research translation program and interviews and focus groups with stakeholders and program users will draw on the context (C), mechanism (M), outcome (O) formula that is core to realist evaluation. In addition, system dynamic methods will capture the feedback loops and complex relationships among the IPT and context-mechanism-outcome configurations. This approach to evaluation of a research translation funding programme may be adapted to similar programmes operating in other settings.

**Ethics and dissemination** Curtin University Human Research Ethics Committee, Western Australia, approved this study (approval number: HRE2020-0464). Results will be published in scientific journals, and communicated to respondents and relevant partners.

## INTRODUCTION

Health systems around the world are facing challenges relating to rising healthcare expenditure and the effectiveness and efficiency of

### STRENGTHS AND LIMITATIONS OF THIS STUDY

⇒ This is the first study to evaluate the impact of the research translation process on health system sustainability.
⇒ Adopting a realist methodology will allow contextual factors and mechanisms to be identified that influence how the research translation process can contribute to health system sustainability.
⇒ A novel feature of this study is the application of system dynamics modelling within the realist evaluation framework, capturing the complexity and feedback relationships between context, mechanism and outcome.
⇒ Using a Delphi method to establish consensus on implementation and evaluation recommendations for a research translation programme will facilitate perspectives to be gained from a broad group of stakeholders.
⇒ Evaluation of one research translation programme as a case example may affect generalisability of study findings to the research translation process more generally.

the system to deliver high value,[1] safe healthcare. A number of complex interdependent factors contribute to these challenges. An ageing population, rising chronic disease, public expectations, and a lack of value-consciousness among healthcare consumers and providers are the major factors driving growing demand for healthcare.[2 3] These issues are occurring in an environment that is resistant to change, often driven by behaviours associated with vested interests, and incentives that do not promote value or transparency.[4] These challenges, pressures and behaviours raise questions about the capacity of the system to deliver affordable, cost-effective outcomes to the population over time, which is often referred to as the sustainability of the system.[5] Research translation is an essential process for ensuring health systems have ongoing capacity to service the health needs of a population and address challenges through the integration

of cost-effective interventions based on new research and technology. While there is a wide range of definitions in the literature in relation to research translation and other associated terminology such as knowledge translation, and research utilisation,[6] a working definition of research translation for the purposes of this study has been adopted from Grimshaw *et al*, as 'ensuring that stakeholders are aware of and use research evidence to inform their health and healthcare decision making'.[7]

Western Australia (WA) is typical of other jurisdictions in Australia facing challenges that threaten the long-term sustainability of its health system. The demand for health services has grown substantially in recent years, along with health expenditure, yet outcomes in population health and acute care in WA have not improved at the same rate.[4] With major health issues such as increasing obesity, an ageing population, chronic disease, mental health and inequalities in health outcomes across the WA population, the sustainability of the WA health system is a growing concern for healthcare leaders and policy makers. The WA Department of Health, along with healthcare leaders, policy makers, clinicians and researchers around the globe, acknowledge the potential of research evidence to improve health outcomes and optimise resource use.[4 8 9] In 2007 the former State Health Research Advisory Council within the WA Department of Health, introduced the Research Translation Projects (RTP) programme.[10] The RTP programme aims to cultivate and translate evidence for a sustainable health system through the support of health service driven, high-quality research projects that have the potential to deliver improved cost-effectiveness and/or efficiencies to the health system while maintaining or improving patient outcomes. Internal evaluations of the RTP programme have focused primarily on research outputs such as the number of publications, changes to practice guidelines and additional research funding obtained. While assessment of research outputs is important, the RTP programme's contribution to health system sustainability is not well understood. Therefore, the protocol presented here aims to outline the theories and methods to explain how the RTP programme supports research translation that in turn contributes to health system sustainability. We provide a rationale for adopting a realist evaluation framework to underpin the evaluation of the RTP programme with the aim of contributing to a more generalisable understanding of the relationships between the process of research translation and its contribution to health system sustainability.

## BACKGROUND AND RATIONALE

Sustainability from a health systems perspective relates to the capacity of the system to be able to service the health needs of the population into the future.[5] It is a multifaceted concept that encompasses both the affordability and efficiency of a system, and the system's ability to adapt and change.[3] Adopting a system thinking approach gives us a framework for understanding the health system as entities, made up of subsystems such as interventions, programmes, organisations, stakeholders and agencies (eg, the emergency departments, quality improvement programmes, safety policies, the workforce, mangers and decision makers and consumers). In this protocol 'health system' means the broader entity as described above. This is an important assertion to make considering the wide-ranging use of the term 'health system sustainability' used in the literature. For example, the term health system sustainability is often used in literature in relation to the discrete subsystems described above (eg, intervention, organisation and agencies). The sustainability of the broader health system, which is the focus here, is not well understood and the question of how does action in the form of multiple research translation activities and projects impact on the sustainability of the wider health system remains unanswered. To address issues that currently threaten health system sustainability, healthcare leaders, policy makers, clinicians and researchers are searching for solutions that capitalise on new research and technologies that promise to deliver safer, value-based care. Such solutions often involve implementing and testing new evidence-based interventions and improvement programmes into the health system, and evaluating their effectiveness, often through pragmatic trials.[11 12]

Increasingly, evaluations also include a focus on the sustainability of the interventions, involving recognition of implementation outcomes such as acceptability, adoption into policy or practice, appropriateness, reach and sustainability to support the spread and adaptation of successful interventions beyond the original setting and context.[11]

As a result, there are a growing number of theories, models and frameworks of sustainability that address individual programmes and interventions.[11 13 14] These theories, models and frameworks provide valuable guidance and insights into the facilitators and barriers that contribute to intervention sustainability.[15] In addition, they provide guidance around funding of interventions that are more likely to continue providing desirable benefits beyond the implementation period usually covered by project funding, as well as guidance for planning and evaluation.[13 14 16–18] The contribution of health improvement interventions to sustainability at the broader system level has received limited attention. This limitation is evident in the measurement of sustainability presented in the literature, which is often presented in terms of continuity of programme activities and outcomes, institutionalisation, adaptation of the intervention components, and sustained attention to the issue or problem.[16 17 19 20]

While sustainability of effective programmes and interventions at the end of the research translation pipeline is valuable, not all interventions need to be sustained in order to be useful or to make a contribution to the larger system goals in which they operate.[17] The process and strategies adopted to facilitate research translation offers several other potential benefits to heath system

sustainability such as workforce collaboration and capacity building, along with the impact that comes from successful implementation.[17 21]

The understanding of research translation as a process within the broader health system, and its contribution to system sustainability needs further exploration. In view of health systems as complex adaptive systems[14 22] the dynamic interaction between and within the system and subsystems means that events at one level of the system influence action and events at another, often in a feedback loop relationship.[23] It follows that, our interest is not solely with individual interventions but rather with the interconnections, interdependencies and feedback loops across a large complex system. While research regarding the key factors that influence the sustainability of interventions is important, the next stage is to move beyond the intervention itself, and investigate how the process of research translation can trigger change within the wider health system.[14 24 25] This requires a shift away from linear cause and effect evaluation design to methods that are capable of dealing with complexity.[26] The process of research translation can be understood as a disruption to the system which is likely to trigger emergent, adaptive behaviours and non-linear, feedback interactions across the system which will either reinforce or diminish overall system sustainability.[26] Therefore, to understand sustainability at the health system level, researchers need to shift their focus from *what* has been achieved, to *how* the process generates change in the system,[27] to identify system traps and opportunities that produce system behaviour. The concept of *how* and *why* change occurs within a system is core to the theory-driven evaluation design, realist evaluation.[27–30]

## Realist evaluation

Realist evaluation is a theory-driven evaluation approach grounded in realism, a philosophy which asserts that both the material and social worlds are real.[28 30] It follows that the reality of the social and material world should be investigated to build a better understanding of what causes change within such systems. Realist evaluation shifts the basic inquiry of evaluation from 'does this work?', to 'what works, in what context and how?'.[28] The realist evaluation approach can be said to be in contrast to traditional linear cause and effect approaches where complexity is controlled for or eliminated.[31] Instead, realist evaluation embraces complexity by acknowledging that context can influence the way an intervention achieves its outcomes through different change mechanisms that are triggered by context. Therefore, the realist approach aims to link intervention to outcome by identifying the various change mechanisms taking place in reality.[28] Within the realist evaluation framework, interventions are 'theories incarnate' embedded in a social reality that makes them prone to interpretation and modification.[32 33] Often these theories are not explicit and exist in the minds of those who designed a particular intervention.[28] It is therefore

a central task of realist evaluation to make the theories explicit in the form of a programme theory.

Programme theories, in their simplest form, depict how a specific intervention is theorised to achieve change.[34] The realist evaluation frameworks seek to develop programme theories as context-mechanism-outcome (CMO) configurations that explain how interventions trigger different change mechanisms across different contexts to achieve (or not achieve) outcomes. The theory is then tested and refined through data collection and analysis that assesses patterns and regularities not only in the programme outputs but also the programme context, the process of implementation, and the mechanisms that may be creating change.[28 29] The outputs from the realist approach lead to a well-articulated programme theory as well as a more generalisable theory, often termed a middle-range theory, falling between programme and grand theory that can be applied across different settings.[34] In the context of health system sustainability, the interest is in the process by which interventions trigger the change process across all levels of the system, from micro intervention level through to the macro policy level of the system. Many parallels can be drawn between the principles underlying realist evaluation and that of systems theory and complexity theory. When adding a system dynamics lens to the evaluation, it becomes clear that the CMO configurations that are central to realist evaluation involve complex, non-linear interactions often in the form of feedback loops. Within complex adaptive systems, the context, mechanism and outcomes are interconnected and, as such, the triggering of one mechanism can impact on the context of another.[35] Realist evaluation is said to be 'methods neutral' and thus provides a perspective and conceptual framework for evaluation rather than strict practical guidelines.[27] This study therefore aims to apply the realist evaluation framework, drawing on system dynamics methods and principles, to explain how research interventions, funded to deliver service improvement and value-based healthcare, impact on sustainability across all levels of a complex adaptive system.

## STUDY AIMS AND OBJECTIVES
### Aim
Produce a programme theory to explain how the research translation process contributes to health system sustainability through the implementation of interventions funded to deliver service improvement and value-based healthcare.

### Objectives
1. Identify the change mechanisms triggered by the implementation of research translation interventions that contribute to health system sustainability,
2. Conduct case study analysis to explore the interconnected and feedback relationships between change mechanisms identified from objective 1, and

3. Refine the Initial Program Theory (IPT) of the impact of the research translation process on health system sustainability based on outcomes from objectives 1 and 2, through validation with key stakeholders to inform a generalisable theory of mechanisms that influence sustainability at a health system level.

## METHODS AND ANALYSIS

A realist evaluation approach is applied to this study.[30] The study draws on a number of theoretical frameworks including complexity science, complex adaptive systems theory and implementation science.[26] It adopts a realist evaluation methodology to examine the pathways by which the RTP programme as a research translation process contributes to health system sustainability. Realist evaluation is increasingly used in health services research to evaluate complex health system interventions.[31 36] This approach has been chosen for its ability to cope with complexity and its focus on understanding how and why complex health interventions trigger change within complex systems. The RAMESES II (Realist And Meta-narrative Evidence Syntheses: Evolving Standards) reporting standards for realist evaluation will be used to structure the reporting of the study methods and analysis.[31] To enhance the trustworthiness of data collection and documentation throughout the project, we will follow Pawson *et al.*[37] Transparency, Accuracy, Purposivity, Utility, Propriety, Accessibility quality standards framework.[38] .

This study will be implemented in three phases, summarised in table 1.

1. IPT development and evidence review: Development and refinement of an IPT for the research translation process and the impact on health system sustainability using peer-reviewed literature, key informant interviews and programme documentation.

2. Programme theory testing: Test the IPT by applying it to selected case studies of interventions funded under the RTP programme using document analysis, key stakeholder interviews and concept mapping workshops with key stakeholders.

3. Theory consolidation: Refine and validate IPT based on the study findings and, peer-reviewed literature where relevant and focus group discussions with relevant stakeholders including programme managers, existing and past investigators of funded projects, and members of the RTP selection panel.

### The RTP programme

The RTP programme, established in 2007 by the WA Department of Health, provides funding for short-term research projects that seek to improve healthcare practice and/or policy in the WA health system. Its aim is to improve or maintain patient outcomes through implementation of RTPs that have the potential to deliver efficiencies to the WA health system. In an environment of rising healthcare burden and limited resources, the RTP programme supports innovation with a focus on high-value care[1] and needs-led research, funding work that might otherwise not have been funded and also providing a focus on issues of importance as identified from the bottom up. Similar programmes such as the US Department of Veteran Affairs' Diffusion of Excellence programme has demonstrated positive results for implementation of evidence-based practice using a bottom-up approach, such as diffusion and sustainability of interventions.[39] In addition to the benefits of bottom-up research, it had been recognised within the WA health system that clinicians were generally uncompetitive for nationally funded grant programmes due to lack of research training and research track record. However, they were ideally placed to identify possible opportunities to implement

**Table 1** The three-phased approach to realist evaluation of research translation in Western Australia

| Phase | Data sources | Analysis |
|---|---|---|
| **Phase I: IPT development and evidence review** | ► Key stakeholder interviews<br>► Programme documentation<br>► Peer-reviewed literature | Qualitative-thematic analysis driven by the realist evaluation and system dynamics principles. |
| Phase II: Programme theory testing | ► Key informant interviews<br>► Document review and analysis of individual RTP reports and other key documents<br>► Secondary data analysis of project data such as cost and outcome data (where necessary) | Iterative process of categorising and connecting strategies, similar to those proposed by Maxwell[53] will be applied to CMO development in light of the information collected during phase II. |
| Phase III: Theory consolidation | ► Key stakeholders' interviews and workshops<br>► Peer-reviewed literature<br>► Delphi Survey | Refine theory that explains how and under which contextual factors research translation brings about health system sustainability.<br>Delphi Survey analysis for consensus relating to recommendation for ongoing research translation practices in the Western Australian health system. |

CMO, context-mechanism-outcome; IPT, Initial Program Theory; RTP, Research Translation Projects.

new practices that promote efficiency while maintaining or improving patient outcomes. The RTP programme was therefore implemented to provide an opportunity to deliver bottom-up innovation and to build research capacity within the clinical environment through enablement and collaboration with those who have experience in research.

The programme is innovative in terms of the requirement for inclusion of health resource appraisal, with a funding criterion that the proposed intervention would reduce or maintain costs to improve health outcomes (or reduce costs and maintain health outcomes). The combination of bottom-up clinician-led research translation, based on front-line need and a key focus on health system efficiency, encourages clinicians from the WA health system to undertake research that has the potential to improve the quality and efficiency of their work. Through research translation into practice, innovation based on front-line need, development of research capacity within the clinical environment, collaboration among research and clinical workforce and the promotion of value-consciousness among clinicians, the RTP programme also aims to contribute to the sustainability of the WA health system. RTPs sit along the research translation continuum and are made up of new research, pilot studies or applications and evaluations of research that have been applied elsewhere.

Given the spread of projects funded by the RTP programme, across multiple settings and addressing a variety of health and health service delivery issues, the programme can be viewed as a microcosm of the broader health system, given a shared focus on making health services more effective and efficient. The RTP programme therefore provides a unique opportunity to study how the process of research translation through clinician-led research can trigger change mechanisms to produce effects that may contribute to sustainability across the WA health system.

### Phase I: IPT development and evidence review

A programme theory describes how the intervention is expected to bring about changes. The IPT is essential to the realist evaluation logic of inquiry.[30] Developing a programme theory shifts the understanding of how an intervention is expected to work from the implicit ideas that often exist in the heads of policy makers, decision makers, clinicians and researchers, to the explicit. According to Luck, a programme theory consists of both a theory of action and a theory of change. Programme theories in terms of the theory of action and expected outcomes can often be identified in policy or programme documents. However in the context of a realist evaluation, further information about the theory of change, that is the mechanism by which the action will achieve the outcomes, is not so obvious and requires the researcher to conduct focused interviews with key stakeholders to elicit the theory of change.[40]

### Data sources

The development of the programme theory will be informed by three main sources: input from the key stakeholders of the RTP programme, analysis of documentation and the literature.[41] Programme stakeholders such as programme managers, existing and past investigators of funded projects, and members of the RTP selection panel (clinicians, health administrators, consumer representatives, health economists) will be interviewed to elicit the hypothesised change mechanisms that explain how the research translation process is thought to influence health system sustainability. In addition, programme documents, including RTP research applications, progress reports and final reports of funded projects as well as past evaluations of the programme will be accessed. While the IPT will be informed and supported by stakeholder interviews, exploration of the literature through a theory-driven review is necessary to make sense of the emerging theories elicited from interviews (ie, hypotheses, hunches, aspirations, intuitions, experiences), inherent to complex interventions applied in heterogeneous contexts.[42]

The literature review will follow the steps of realist synthesis review outlined by Pawson *et al*[43] and Jagosh[44] to identify published theories of change within the context of the research translation process (eg, theories focusing on change within the individual professionals, within the organisation, the social setting or economic context).[45]

### Data analysis

The CMO framework will underpin an iterative process of reflection and adaptation, which will take place to analyse data and identify relations between contexts, mechanisms of change and outcomes for analysis of stakeholder interviews and literature review. The emergent behaviour of complex adaptive systems such as the health system would suggest that feedback loops would exist between CMOs; therefore, the IPT will be depicted using a causal loop diagram (CLD) to capture any feedback interactions between CMOs. While logic models are commonly used to depict a programme theory, the complexity of both the intervention and the system in which it is being implemented lends itself to system dynamics methods to depict the programme theory.

### Phase II: Program theory testing and validation

We will test the programme theory of the research translation process developed in phase I by analysing in detail a series of RTP programme case studies. Realist evaluation aims to test programme theories for the purpose of refining them as well as informing implementation of the programme. As such the core questions asked of the programme are; what works (and does not work) for whom, in what circumstances and how? Mechanisms that produce positive outcomes in one context may not produce the same outcomes in an alternative context. This notion that the success (or failure) of an intervention is context-dependent is one shared in the current literature that explores implementation science and

complex interventions.[26] Anderson and Hardwick outline this concept succinctly:

'Recognising that no one policy, programme or intervention will always work, all the time, for everyone, realist approaches seek to explain this pattern of outcomes through building programme theories about how an intervention (policy or programme) is meant to work (often according to programme architects, or policymakers, or participants), and then "test" whether and how this programme theory plays out in the real world using empirical data.'[46] The question that is at the core of the application of realist evaluation proposed here is what happens to the individual projects funded by RTP and are there common system pitfalls or opportunities that lead to their success or failure in terms of research translation.

Selection criteria: Given the variety of implemented research projects funded by the RTP programme, a descriptive analysis of past RTP projects will be undertaken prior to a case study selection to enable a selection of cases that captures the variety of activity undertaken within the RTP programme. Projects will be classified based on characteristics such as context of implementation (eg, metro hospital, rural hospital, primary care, community health) and type of research translation intervention (eg, new practice guideline, new service, new test, role substitution).[11] In the first instance, case study selection will be purposive in nature by categorising projects into a matrix of the above criteria to ensure a breadth of projects across those criteria is captured. However, given the retrospective nature of this study, case study selection may be limited by access to project investigators and other stakeholders for interview.

## Data sources

Case study analysis will be undertaken using three data collection techniques: key informant interviews, document review and analysis of individual RTP reports and other key documents, and secondary data analysis of project data such as cost and outcome data where necessary.

## Data analysis

Qualitative data, including documentation, interviews and focus groups, will be analysed in NVivo using the CMO configuration as a guide for analysis. Several authors highlight the absence of guidance within the realist evaluation approach as to specific analytical tools to be used.[47–50] The qualitative data from the case studies will undergo an iterative process of thematic analysis by which preliminary codes will be developed for themes identified in the IPT and more specific codes and themes being induced from further cycles of thematic analysis. This will also involve a process of mapping codes against the CLD developed in phase I. The IPT will then be refined to best reflect the emerging findings. The aim of this phase in the realist approach is to identify emerging patterns across the case studies to validate and refine the IPT to be consolidated in phase III. In addition, the interactions and relationships between change mechanisms will be refined.

## Phase III: Theory consolidation

The final phase of research will involve refining the IPT to produce a final iteration of the research translation process to be presented in the form of a mid-range programme theory of its impact on sustainability of the system. The final model will articulate a model of complex relationships among programme and health service unit, organisation and policy-level processes of research translation for sustainability of the health system. Like the process undertaken in phase I, the theory consolidation will be informed through the input of key stakeholders of the RTP programme and a rapid review of the literature to update the initial realist review to include relationships and interconnections between change mechanisms. Input from key stakeholders will be elicited through concept mapping workshops and interviews where the theory will be presented to key stakeholders for validation.

## Implementation and evaluation

In addition to the refined research translation theory, a Delphi Survey of experts will be undertaken to determine implementation and evaluation recommendations for research translation within the WA health system into the future. The survey will be conducted across several rounds via online questionnaire, which will be delivered until consensus is reached. The Delphi panel will consist of those involved in delivering the RTP programme at the WA Department of Health, investigators involved in individual projects and other stakeholders.[51 52] These recommendations will include an evaluation framework specifically for the RTP programme to be addressed by future funding recipients and selection panels.

**Contributors** AM, DH, EG, YZ, SR jointly conceived the study; AM and DH developed the study proposal; AM led the writing of this paper; DH, EG, YZ and SR read and reviewed the final version prior to submission.

**Funding** This work is supported by the National Health and Medical Research Council Partnership Centre for Health System Sustainability (ID: 9100002). In addition the Western Australian Department of Health provides financial support for the Research Translation Projects programme.

**Competing interests** None declared.

**Patient and public involvement** Patients and/or the public were not involved in the design, or conduct, or reporting, or dissemination plans of this research.

**Patient consent for publication** Not applicable.

**Provenance and peer review** Not commissioned; externally peer reviewed.

**Data availability statement** Data sharing is not applicable as no data sets were generated and/or analysed for this study.

**ORCID iDs**
Abby Mosedale http://orcid.org/0000-0002-4089-725X
Delia Hendrie http://orcid.org/0000-0001-5022-5281
Yvonne Zurynski http://orcid.org/0000-0001-7744-8717
Suzanne Robinson http://orcid.org/0000-0001-5703-6475

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
