## [Reviewer comments · BMJ Open]

ARTICLE DETAILS

TITLE (PROVISIONAL)	Realist evaluation of the impact of the research translation process on health system sustainability: a study protocol
AUTHORS	Mosedale, Abby; Hendrie, Delia; Geelhoed, Elizabeth; Zurynski, Yvonne; Robinson, Suzanne

VERSION 1 – REVIEW

REVIEWER	David Goodrich US Department of Veterans Affairs
REVIEW RETURNED	01-Jan-2021

GENERAL COMMENTS	In a dynamic and rapidly evolving health care environment, there is urgency to rapidly translate new evidence-based interventions into clinical practice for sustained impact at the service, facility and system levels. The current protocol proposes to develop a program theory using realist methodology to explain how the research translation process contributes to health system sustainability through the implementation of interventions at multiple levels of impact using contributions from Western Australia's Department of Health's Research Translation Projects (RTP) program. While this manuscript is a protocol, it is not without a number of major and minor issues for clarification: Major: 1. Major Study Strengths and Limitations, p. 3. While the authors mention the use of systems modeling techniques and the Delphi method to establish end-user consensus as paper strengths, there is a lack of detail on the specific protocol methods that the researchers will use to carry out this section of the protocol.2. Page 3, line 32. It would be helpful for readers for the authors to define what they mean by high value (vs. low value services).3. Page 3, line 39. Please specify what type of cultural backdrop is resistant to change (racial, socioeconomic, institutional, providers, etc.).4. Page 3, line 46. Please provide an operational definition of research translation for readers.5. Page 4, line 27. Without context and definition, it is unclear what "adopting a realist evaluation framework" does for this protocol. Consider omitting here unless explanation is provided.6. Page 4, lines 46-47. Please clarify what is meant by "strategies" here.7. Page 4, lines 49-50. Please clarify what is meant by research translation interventions. There is ambiguity in this paragraph about "what" is being implemented and "how" it is being implemented (i.e., implementation strategies) as well as how these two constructs interact to affect sustained implementation outcomes. Readers may benefit by providing a working definition of the range of innovations
---

	that constitute a “cost effective intervention.” 8. Page 4, line 52 – page 5, line 25. This section lacks conceptual clarity regarding the term sustainability. Specifically, the authors seem to be discussing three distinct levels of evaluation but never clearly define sustainability for each of the 3 levels. Moreover, the definition of spread, outcomes, and “sustained implementation” are unclear. An intervention that has sustained use by definition has been implemented and integrated into routine processes of care until otherwise, de-implemented. 9. Page 4, lines 53-54. The authors reference evidence-based policy making and evaluation yet nowhere in this report do the authors specify the methods or designs needed for strong program evaluation to evaluate program impact across settings and time. What is the role of randomization in ensuring results from program implementations are externally valid and generalizable as spread occurs? 10. Page 5, lines 6-12. There is a glaring omission of relevant theories related to program adaptation by Stirman and colleagues as well as dynamic sustainability of interventions over time (DSF, Adaptome by Chambers and colleagues; Aarons et al. Dynamic Adaptation Process). The authors are encouraged to cite these. 11. Page 5, line 20. The authors note that not all interventions need to be sustained; it would be appropriate to accurately describe the need for planned deintensification or de-implementation of low value services here. 12. Page 5, lines 27-29. Is this research evaluation gap true only in Australia? It does not seem to be the case in the US, Canada, Taiwan, and many northern European countries where national evaluations of implementation interventions are conducted and reported regularly for regional and national health decisionmakers. 13. Page 5, lines 51-53. The authors are encouraged to examine and reference Mary Ann Scheirer’s work (2005, 2008, 2011) on defining and evaluating program sustainability and, rephrasing this assertion. 14. Page 6, line 58 – page 7, line 8. It is unclear from the “realist” perspective when it becomes unrealistic to test an intervention across multiple settings to enhance generalizability for diverse contexts. Likewise, when do CMO configuration analyses have sufficient power or sample size to make develop a robust program theory? This section also largely omits the large body of literature on rapidly and iteratively learning as an intervention is spread to other contexts to make adaptations for local contexts to improve program fit while retaining fidelity to the program’s core logic. It is also unclear in this section regarding what statistical or ethnographic methods are used to establish CMO configurations and feedback loops and who performs these analyses (researchers, evaluators?). 15. Page 8, Methods and Analysis. Compared to most protocol papers, there is a general lack of specificity regarding timelines of data collection, who will be collecting data when and from how many people using which specific methods. Descriptions of data and analyses are equally opaque and make it hard for reviewers, researchers, and general leaders to know the specific protocol steps were carried out with fidelity (or not) to help interpret later outcome papers. The broad methods descriptions also make replication challenging which is a hallmark of strong scientific reports. 16. Page 9, line 10. It would be appropriate to compare how RTP is similar to other programs promoting bottom-up engagement of providers such as the US Dept of Veteran Affairs’ Diffusion of Excellence/Innovation Ecosystem or initiatives by the US NIH/National Cancer Institute.
--	---

	17. Page 10, line 16. What disciplines are represented on the RTP selection panel? (implementation scientists, economists, behavioral medicine experts, quality improvement engineers, public health researchers?) 18. Page, 10, line 23, “rapid review” This is a large body of literature in health and medicine with an even larger database from organizational management and change literature in business, agriculture, and education. How will the relevant disciplines be scanned rapidly? 19. Page 10, line 36-37. This description is unclear. Generally, very specific quantitative methods are used to establish such causal relationships. If these causal models are being proposed here, it seems appropriate to mention which statistical methods will be used to test the feedback interactions within the framework. 20. Page 10, line 48. Are the researchers "testing" the process or in fact validating it for concurrent or face validity with stakeholders? 21. Page 10, line 55. There is a curious omission of the relevant literature from implementation science on the tension between fidelity to "core" program elements and the necessity of "adapting" modifiable program elements to improve fit between intervention, implementation strategy and context. How will causal mechanisms of contextual modification be tested in a rigorous manner? 22. Page 11, 40-41. CFIR does not provide a framework to evaluate outcomes or mechanisms of change. CFIR provides a determinants framework to describe contextual barriers and facilitators that may be correlated with outcomes and possible mechanisms of change. 23. Page 12, line 3. No quantitative analyses have been described. How will the final model be robust and “empirically” tested? 24. Page 12, line 17. The description of the methods pertaining to conducting and analyzing the findings from the Delphi Panel lack detail. 25. Conclusion? Minor 1. Page 3, lines 6-7. This is not the first manuscript on the impact of research translation on sustainability. For example, the U.S. Department of Veterans Affairs QUERI program just proposed a framework (Braganza, Kilbourne, JGIM; 2020) evaluating key domains of their translational research program on system outcomes. Moreover, the VA has a Research to Real World use framework design to facilitate knowledge translation across its entire translational research pipeline. These frameworks parallel similar work that has been conducted in Canada, the US AHRQ, and the PCORI Institute over the past two decades. 2. Page 4, line 13. Does “industry-driven” refer to innovations sponsored by corporations, or to interventions inspired by health care workers and the regional health care systems? This section is unclear as to the purpose of the RTP; is it meant to help researchers get their programs into practice or is it meant to improve the quality and safety of services? If the latter, why are meaningful clinical outcomes not currently tracked by RTP projects?
--	--

REVIEWER	Isomi Miake-Lye VA Greater Los Angeles HSR&D Center of Excellence
REVIEW RETURNED	16-Jan-2021

GENERAL COMMENTS	I enjoyed reading this protocol and look forward to the results of this work, which have the potential to fill an important gap in the existing conversation by focusing on the system level sustainability. The
--

manuscript was well written and compelling.

VERSION 1 – AUTHOR RESPONSE

Reviewer 1 comment	Author response
Reviewer: 1 Dr. David Goodrich, US Department of Veterans Affairs Comments to the Author: In a dynamic and rapidly evolving health care environment, there is urgency to rapidly translate new evidence-based interventions into clinical practice for sustained impact at the service, facility and system levels. The current protocol proposes to develop a program theory using realist methodology to explain how the research translation process contributes to health system sustainability through the implementation of interventions at multiple levels of impact using contributions from Western Australia’s Department of Health’s Research Translation Projects (RTP) program. While this manuscript is a protocol, it is not without a number of major and minor issues for clarification:	Thank you very much for taking the time to provide such detailed and valuable feedback on our study protocol. We have responded to each of the comments and queries raised, and have amended the manuscript accordingly to improve clarity. Please note that page and line numbers correspond to manuscript with track changes included rather than clean copy of transcript.
Major	
1. Major Study Strengths and Limitations, p. 3. While the authors mention the use of systems modelling techniques and the Delphi method to establish end-user consensus as paper strengths, there is a lack of detail on the specific protocol methods that the researchers will use to carry out this section of the protocol.	Page 15, line 15 Additional references and some detail have been included within the methods section of the protocol to direct the reader to established methods for Delphi study and system dynamics modelling which will be applied to this study. (Boukdedid, Abdoul, Loustau, Sibony, & Alberti, 2011; Hasson, Keeney, & McKenna, 2000). This is a complex study with multiple components and we have attempted to portray the methods within the word limit constraints of the BMJ Open.
2. Page 3, line 32. It would be helpful for readers for the authors to define what they mean by high value (vs. low value services).	Page 3, line 16. A citation has been included here (Elshaug et al., 2017)
3. Page 3, line 39. Please specify what type of cultural backdrop is resistant to change (racial, socioeconomic, institutional, providers, etc.)..	Page 3, line 20. The terminology has been changed from cultural backdrop to ‘an environment’.
4. Page 3, line 46. Please provide an operational definition of research translation for readers.	Page 3, line 27 – 31 A citation has been included (Grimshaw, Eccles, Lavis, Hill, & Squires, 2012) as well as a working definition for research translation. “...a working definition of research translation for the purposes of this study has been adopted from Grimshaw and colleagues, as “ensuring that stakeholders are aware of and use research evidence to inform their health and healthcare decision making”
5. Page 4, line 27. Without context and definition, it is unclear what “adopting a realist evaluation framework” does for this protocol.	Page 4, line 19 The realist evaluation framework is the overall approach that will be taken for the evaluation. The sentence has been reworded to

Consider omitting here unless explanation is provided.	express this more clearly.
6. Page 4, lines 46-47. Please clarify what is meant by “strategies” here.	Page 5, line 26 – 29. The term ‘strategies’ has been replaced by ‘solutions’ in conjunction with additional text, “ ...solutions that capitalise on new research and technologies that promise to deliver safer, value-based care. Such solutions often involve implementing and testing new evidence-based interventions and improvement programs into the health system, and evaluating their effectiveness, often through pragmatic trials”
7. Page 4, lines 49-50. Please clarify what is meant by research translation interventions. There is ambiguity in this paragraph about “what” is being implemented and “how” it is being implemented (i.e., implementation strategies) as well as how these two constructs interact to affect sustained implementation outcomes. Readers may benefit by providing a working definition of the range of innovations that constitute a “cost effective intervention.”	Page 4, line 25 – page 7, line 25 The paragraph titled ‘Background and Rational’ has been reworded to address an overall lack of clarity that was made apparent from the comments of reviewer 1. The change to the term strategies is part of larger changes to the paragraph which aims to depict the system, rather than intervention focus of the research. Text has been changed within this paragraph to clarify what we mean by implementation outcomes – our study is focussed on research translation which includes the implementation of research evidence into the healthcare system and policy.
8. Page 4, line 52 – page 5, line 25. This section lacks conceptual clarity regarding the term sustainability. Specifically, the authors seem to be discussing three distinct levels of evaluation but never clearly define sustainability for each of the 3 levels. Moreover, the definition of spread, outcomes, and “sustained implementation” are unclear. An intervention that has sustained use by definition has been implemented and integrated into routine processes of care until otherwise, de-implemented.	Page 4, line 25 – page 7, line 25 The Background and Rational paragraph has been substantially re-written to improve clarity while providing definitions and key references where possible. In terms of “sustained implementation” the point has been noted and text reworded to reflect the reviewer’s comment that “An intervention that has sustained use by definition has been implemented and integrated into routine processes of care until otherwise, de-implemented.” Along with other changes to the paper, text has been added to improve the clarity of this assertion and the overall focus of the paper.
9. Page 4, lines 53-54. The authors reference evidence-based policy making and evaluation yet nowhere in this report do the authors specify the methods or designs needed for strong program evaluation to evaluate program impact across settings and time. What is the role of randomization in ensuring results from program implementations are externally valid and generalizable as spread occurs?	Page 5, line 27-29. We would like to acknowledge that new evidence-based interventions and improvement programs in the health system are evaluated through pragmatic trials. The text has been changed to reflect this in the background and rational of the protocol paper. It follows that the specific methods and design needed for strong program evaluation is perhaps not applicable here given that the overall aim of the study is not to evaluate the impact of implementation

	of the RTP program across multiple settings and time, nor the individual projects funded by the RTP program. Rather the aim is to understand the research translation process within the system as a contributing factor to health system sustainability. A significant amount of text has been changed to better reflect this (see background and rational). The RTP Program is a case study being used to understand what happens to individual projects(interventions). Some outcomes may be that they continue in the same setting, adapt and spread to others or cease to operate after funding has been withdrawn. The combination of realist evaluation approach, systems thinking perspective and case study analysis will be adopted to understand the mechanisms that lead to various outcomes that can only be Page 4 of 10 determined when case studies are undertaken. Overall, this will give us a picture of how research translation is occurring in the WA health system and how we can avoid undesirable feedback loops and exploit opportunities for more effective research translation.
10. Page 5, lines 6-12. There is a glaring omission of relevant theories related to program adaptation by Stirman and colleagues as well as dynamic sustainability of interventions over time (DSF, Adaptome by Chambers and colleagues; Aarons et al. Dynamic Adaptation Process). The authors are encouraged to cite these.	We acknowledge that program adaptation is an important aspect of intervention sustainability, and the resources suggested by the reviewer will certainly be considered given that theories relating to program adaptation will likely be a phenomenon observed as part of the case study analysis.
11. Page 5, line 20. The authors note that not all interventions need to be sustained; it would be appropriate to accurately describe the need for planned deintensification or de-implementation of low value services here.	The de-implementation or planned deintensification is a potential finding of the case studies. If this is an observed finding of projects, we will aim to explore further to understand the reasons for deintensification and disinvestment. The note about not all interventions needing to be sustained to contribute to health system sustainability is to highlight that the research translation process may have benefits to whole of system sustainability regardless of the translated research being sustained in the form of an intervention.
12. Page 5, lines 27-29. Is this research evaluation gap true only in Australia? It does not seem to be the case in the US, Canada, Taiwan, and many northern European countries where national evaluations of implementation interventions are conducted and reported regularly for regional and national health decisionmakers.	Evaluation of implementation interventions at the intervention level (subsystem or micro level) is reported on regularly, however not at the whole of system level. We have attempted to clarify what is meant by the concept of a 'whole of system level' in the Background and Rational section of the protocol.
13. Page 5, lines 51-53. The authors are encouraged to examine and reference Mary Ann Scheirer's work (2005, 2008, 2011) on	Page 6, line 4. Citation added elsewhere. This assertion remains and has been made with more clarity around the system focus of this research.

defining and evaluating program sustainability and, rephrasing this assertion.	
14. Page 6, line 58 – page 7, line 8. It is unclear from the “realist” perspective when it becomes unrealistic to test an intervention across multiple settings to enhance generalizability for diverse contexts. Likewise, when do CMO configuration analyses have sufficient power or sample size to make develop a robust program theory? This section also largely omits the large body of literature on rapidly and iteratively learning as an intervention is spread to other contexts to make adaptations for local contexts to improve program fit while retaining fidelity to the program’s core logic. It is also unclear in this section regarding what statistical or ethnographic methods are used to establish CMO configurations and feedback loops and who performs these analyses (researchers, evaluators?).	Page 9, line 25-27 The research data used to develop the CMO configurations is qualitative and viewed from a realism lens, as such the traditional approaches to validity and generalisability such as sample size, power and statistical methods for analysis are viewed differently here. Drawing on the original theory of Pawson and Tilley, through the realist lens, validity is related to the degree to which the researcher has included the multiple perspectives pertaining to a given situation. When applying a realism philosophy to evaluation research the major aspect of promoting validity is to acknowledge that programmes or interventions will be viewed differently from the different perspectives of the multiple stakeholders. Different stakeholders will have a different perspective on the programs effectiveness and factors that promote or inhibit that effectiveness. We will follow Pawson, Boaz, Grayson, Long, and Barnes (2003) Transparency, Accuracy, purposively, Utility, Propriety, Accessibility and Specificity (TAPUPAS) criteria to enhance the trustworthiness of data collection and documentation (Flynn, Rotter, Hartfield, Newton, & Scott, 2019). The TAPUPAS quality standard framework has been included in the protocol paper. In a realist evaluation and in view of drawing transferable lessons about the research translation process, a realist analysis to identify the relationship between context mechanism and outcome is applied. (Pawson and Tilley 1997). In a realist analysis, this relationship is not seen as fixed, rather, it is interpreted as certain conditions creating generative or conditional causality. To unpack generative causality, an iterative theory testing strategy will be used, the initial program theory and proposed ‘theories of change’ are examined for their utility and efficacy across a various context (the individual research translation projects, across different services, wards and settings). Data analysis strategies of categorising and connecting, similar to those proposed by Maxwell(Maxwell, 2012) will be applied to CMO development.
15. Page 8, Methods and Analysis. Compared to most protocol papers, there is a general lack of specificity regarding timelines of data collection, who will be collecting data when and from how many people using which specific methods. Descriptions of data and analyses are equally opaque and make it hard for reviewers, researchers, and general leaders to know the	Page 10, line 12 Data collection will be undertaken by the research team. A table has been added to summarise the steps outlined in phases 1-3. The publication reporting the findings at each phase will report the methods in more detail.

specific protocol steps were carried out with fidelity (or not) to help interpret later outcome papers. The broad methods descriptions also make replication challenging which is a hallmark of strong scientific reports.	
16. Page 9, line 10. It would be appropriate to compare how RTP is similar to other programs promoting bottom-up engagement of providers such as the US Dept of Veteran Affairs' Diffusion of Excellence/Innovation Ecosystem or initiatives by the US NIH/National Cancer Institute.	Page 11, line 9 – 12. Thank you for highlighting these programs which are indeed similar in the bottom up approach to research translation and implementation. Citation and acknowledgement has been included.
17. Page 10, line 16. What disciplines are represented on the RTP selection panel? (implementation scientists, economists, behavioural medicine experts, quality improvement engineers, public health researchers?)	Page 12, line 20-21. The selection panel is made up of clinicians, consumers, health administrators and public health researchers. This has been added to the protocol paper.
18. Page, 10, line 23, "rapid review" This is a large body of literature in health and medicine with an even larger database from organizational management and change literature in business, agriculture, and education. How will the relevant disciplines be scanned rapidly?	Page 12, line 29 - 32 We take note of this comment and consider the appropriateness of the term rapid review. We will apply a realist synthesis approach to better align with the realist philosophy. Realist synthesis can be paralleled with the process of an evidence check, often undertaken by organisation such as the Sax Institute to develop a concise summary of relevant literature. The details of the realist review methodology has been cited in the protocol and will be outlined in more detail in future publications.
19. Page 10, line 36-37. This description is unclear. Generally, very specific quantitative methods are used to establish such causal relationships. If these causal models are being proposed here, it seems appropriate to mention which statistical methods will be used to test the feedback interactions within the framework.	The purpose of the CLD is to capture the dynamic cycles of influence that would serve to pinpoint where leverage points (system traps and opportunities) in the system exist. This is explored further in phase 2, program theory testing where analysis involves an iterative process of 'back and forth' between the initial program theory data collected in phase 2 (Flynn et al., 2019). The development of a CLD itself does not include statistical methods however for the construction of a stock-and flow diagram leading to a simulation model, further quantitative detail is needed that will allow model builders to convert the CLD into a SFD (stock-and flow diagram). For this study we are not attempting to develop a stock and flow model, instead just a qualitative representation of the relationships in the form of a CLD.
21. Page 10, line 55. There is a curious omission of the relevant literature from implementation science on the tension between fidelity to "core" program elements and the necessity of "adapting" modifiable program elements to improve fit between intervention, implementation strategy and context. How will	Please see response to comments 14 and 20.

causal mechanisms of contextual modification be tested in a rigorous manner?	
22. Page 11, 40-41. CFIR does not provide a framework to evaluate outcomes or mechanisms of change. CFIR provides a determinants framework to describe contextual barriers and facilitators that may be correlated with outcomes and possible mechanisms of change.	Page 14, line 21 This is an important oversight that reviewer 1 has highlighted and we appreciate the guidance here. This has been omitted from the protocol.
23. Page 12, line 3. No quantitative analyses have been described. How will the final model be robust and “empirically” tested?	Page 14, line 33. An important point highlighted by reviewer 1. The term empirically tested has been removed.
24. Page 12, line 17. The description of the methods pertaining to conducting and analyzing the findings from the Delphi Panel lack detail.	Page 15, line 15 Also addressed in comment 1. Page 8 of 10 See citation inserted to standard methods for Delphi which will be adopted here. This has not been expanded due to the constraints on the word allowance. (Boukdedid et al., 2011; Hasson et al., 2000)
25. Conclusion?	Other protocol papers in BMJ open have concluded with the discussion. Was not included separately due to the word limitations.
Minor	
1. Page 3, lines 6-7. This is not the first manuscript on the impact of research translation on sustainability. For example, the U.S. Department of Veterans Affairs QUERI program just proposed a framework (Braganza, Kilbourne, JGIM; 2020) evaluating key domains of their translational research program on system outcomes. Moreover, the VA has a Research to Real World use framework design to facilitate knowledge translation across it's entire translational research pipeline. These frameworks parallel similar work that has been conducted in Canada, the US AHRQ, and the PCORI Institute over the past two decades.	To our knowledge this is the first study to look at the impact of research translation of the sustainability of the system as a whole complex adaptive system.
2. Page 4, line 13. Does “industry-driven” refer to innovations sponsored by corporations, or to interventions inspired by health care workers and the regional health care systems? This section is unclear as to the purpose of the RTP; is it meant to help researchers get their programs into practice or is it meant to improve the quality and safety of services? If the latter, why are meaningful clinical outcomes not currently tracked by RTP projects?	Page 4, line 10 The term health service driven has been used instead of industry driven to be more specific.
Reviewer 2 comments	
Ms. Isomi Miake-Lye, VA Greater Los Angeles	Thank you for taking the time to review this study

HSR&D Center of Excellence, UCLA Fielding School of Public Health Comments to the Author: I enjoyed reading this protocol and look forward to the results of this work, which have the potential to fill an important gap in the existing conversation by focusing on the system level sustainability. The manuscript was well written and compelling.	protocol. We appreciate your enthusiasm for the study and look forward to publishing its results.
---	--